# Condemnation of Porcine Carcasses: A Two-Year Long Survey in an Italian High-Throughput Slaughterhouse

**DOI:** 10.3390/vetsci10070482

**Published:** 2023-07-24

**Authors:** Alfonso Rosamilia, Giorgio Galletti, Stefano Benedetti, Chiara Guarnieri, Andrea Luppi, Stefano Capezzuto, Marco Tamba, Giuseppe Merialdi, Giuseppe Marruchella

**Affiliations:** 1Istituto Zooprofilattico Sperimentale della Lombardia e dell’Emilia-Romagna “Bruno Ubertini” (IZSLER), 25124 Brescia, Italy; giorgio.galletti@izsler.it (G.G.); andrea.luppi@izsler.it (A.L.); marco.tamba@izsler.it (M.T.); giuseppe.merialdi@izsler.it (G.M.); 2Local Health Unit Authority, 41121 Modena, Italy; st.benedetti@ausl.mo.it (S.B.); c.guarnieri@ausl.mo.it (C.G.); 3Local Health Unit Authority, 43125 Parma, Italy; 4Department of Veterinary Medicine, Località Piano d’Accio, University of Teramo, 64100 Teramo, Italy; gmarruchella@unite.it

**Keywords:** pigs, slaughterhouse, ante-mortem inspection, post-mortem inspection, carcass condemnation

## Abstract

**Simple Summary:**

The condemnation of whole carcasses after ante-mortem or post-mortem inspection represents the failure of a long period of challenging work, and it is regarded as a highly undesirable event. Therefore, the estimation of carcass condemnation is valuable for preserving both consumers’ health and pig farming profitability. The present survey aimed to report data about the condemnation of porcine carcasses in an Italian high-throughput slaughterhouse over 2 years. A total of 2,062,278 pigs were slaughtered during the study period. Overall, 1362 pigs were considered unfit for slaughtering after ante-mortem inspection, mostly because of death during transportation or in resting pens, after their arrival at the abattoir. Moreover, 2007 carcasses were condemned after post-mortem inspection, mostly due to diffuse peritonitis, disseminated abscesses, slaughter chain faults, and erysipelas. The critical analysis of the data underlines the need to further standardize ante-mortem and post-mortem judgements, in order to best exploit the abattoir as a useful source of epidemiological information.

**Abstract:**

Ante-mortem and post-mortem inspection at slaughter are performed by Official Veterinarians and it is essential to identify alterations/lesions, which can make organs/carcasses unsuitable for human consumption. Obviously, carcass condemnation must be regarded as a highly undesirable event for the entire swine industry chain, as it represents the total failure of a long period of challenging work. Therefore, it seems valuable to estimate the prevalence and causes of carcasses condemnation, in order to preserve consumers’ health and pig farming profitability. Bearing that in mind, the present study aimed at providing a reliable picture of the condemnation of porcine carcasses in Italy, with a special emphasis being placed upon pathological findings. Data were collected in a high-throughput abattoir located in northern Italy, where a total of 2,062,278 pigs were slaughtered during the period of study (2021–2022). Overall, 1362 pigs were considered unfit for slaughtering after ante-mortem inspection, mostly because of death during transportation or in resting pens, after their arrival at the abattoir. Moreover, 2007 carcasses were condemned after post-mortem inspection. The most common causes of condemnation were severe and diffuse peritonitis, disseminated abscesses, jaundice, and erysipelas. In our opinion, the present survey may provide useful and updated information about the condemnation of pig carcasses in Italy. At the same time, it highlights the need to collect data in a more systematic and standardized way, thus making possible their comparison over time and among different geographic areas.

## 1. Introduction

The systematic collection and sharing of data along the entire food chain (“from farm to fork”) is crucial for modern meat inspection, even to prevent zoonoses through a One Health multidisciplinary approach [1,2]. For this reason, the slaughterhouse (also referred to as an abattoir in the present work) holds a strategic position, acting as an irreplaceable epidemiological control point. European legislation defines the slaughterhouse as “an establishment used for slaughtering and dressing animals, the meat of which is intended for human consumption” [3]. Moreover, it also represents a relevant source of useful epidemiological information, provided by Official Veterinarians (OVs) during meat inspection [4,5]. Meat control—developed to protect consumers from foodborne hazards, as well as to ensure food safety and quality—is mainly based on ante-mortem and post-mortem inspections performed by OVs at slaughter [4]. More recently, meat inspection activities have broadened their aims to fulfil several tasks from a real One Health perspective: (i) verifying compliance with human health, animal health, and animal welfare requirements; (ii) controlling animal by-products; and (iii) protecting the environment [5].

Ante-mortem inspection (AMI) should take place within 24 h after the arrival of pigs at the slaughterhouse and it is intended to verify whether animal health and welfare have been compromised, to recognize any condition making fresh meat unfit for human consumption (including zoonotic and epizootic diseases), and to detect any evidence of the use of prohibited/unauthorized substances, the misuse of veterinary drugs, or the presence of chemical residues or contaminants [6]. Based on the results of the AMI, pigs may be considered fit for slaughter. Alternatively, slaughter may be postponed and/or additional procedures may be decided at post-mortem inspection (PMI) and/or the carcasses may be culled and destroyed [6,7]. It is noteworthy that Ghidini et al. [7] have recently shown that AMI findings can be useful for predicting specific disease conditions found at PMI, thus contributing to the classification of pig farms following a risk-based approach. In more detail, dirtiness and skin lesions were demonstrated to be correlated with pneumonia, pleurisy, skin wounds, dermatitis, and kidney lesions, while lameness was correlated with the presence of abscesses.

Post-mortem inspection provides an invaluable tool with which to determine whether meat is suitable for human consumption, through the detection of lesions on carcasses and pluck. In the European Union (EU), the visual inspection of porcine carcass and offal is prescribed at PMI, with additional examinations (e.g., palpation, incision, laboratory tests) being performed when indicated (Article 14 of Commission Implementing Regulation (EU) 2019/627) [6]. Moreover, PMI represents a very efficient point of surveillance from which to assess animal health. Data collected at PMI are of paramount relevance to implementing and monitoring the effect of animal health plans—this being particularly true along the pork chain. As a matter of fact, a growing body of evidence indicates that farmers and veterinarians pay attention to the reports from the slaughterhouse (e.g., dealing with pneumonia and pleurisy scores), which reasonably stimulates a more effective and efficient management of diseases [8].

European legislation lists all the cases in which OVs shall declare fresh meat unfit for human consumption. Among those cases, some conditions are detectable at PMI, e.g., animals emaciated or affected by a generalized disease (i.e., systemic septicemia, pyemia, toxemia or viraemia), showing pathological or organoleptic changes (i.e., a pronounced sexual odor or insufficient bleeding), soiling, fecal, or other contamination, exhibiting parasitic infestation, or containing foreign bodies (Article 45 of Commission Implementing Regulation (EU) 2019/627) [6]. Obviously, the condemnation of whole carcasses is always regarded as a highly undesirable event, as it represents the total failure of a long period of challenging work. Therefore, it seems valuable to estimate the prevalence and causes of carcasses condemnation in order to preserve both consumers’ health and pig farming profitability.

## 2. Materials and Methods

The present study was carried out between January 2021 and December 2022 in a high-throughput slaughterhouse located in the Emilia-Romagna region (Northern Italy), where approximately 1,000,000 pigs per year are slaughtered (about 10% of total pigs slaughtered in Italy). Most of the slaughtered pigs are “heavy pigs” (i.e., at least nine months old, with a body weight of about 160–170 kg), which are intended for Protected Designation of Origin (PDO) “Parma Ham” production (Consortium for Parma Ham 1992).

As a routine, OVs recorded all pigs unfit for slaughtering after AMI, as well as total carcasses condemnation after PMI, according to Regulation (UE) 2019/627 [6]. In particular, the following information was detailed in a Microsoft^®^ Excel sheet: date of slaughtering, batch number, number of pigs per batch, identification code of the pig herd of origin, and cause of condemnation.

Descriptive statistics about the causes of pigs unfit for slaughtering after ante-mortem inspection and post-mortem carcass condemnations were provided. The Pearson’s Chi-squared test was used to evaluate the effect of the season, if any, on carcass condemnation. Analyses were performed using R 4.0.2 software.

## 3. Results

During the study period, a total of 2,062,278 pigs were slaughtered (mean value 4473 slaughtered pigs/working day), belonging to 1380 herds and 17,442 batches (mean value 38 batches/working day).

Overall, 1362 pigs were considered unfit for slaughtering after AMI, mostly because of death during transportation or in resting pens, after their arrival at the abattoir. Moreover, some pigs were euthanized (emergency culling) in order to spare them from unresolvable painful and/or distressful conditions detected at AMI, such as severe dyspnea or prolonged lateral recumbency (see Table 1 for details).

A total of 2007 carcasses were condemned after PMI (see Table 2 for details). Main pathological findings are shown in Figure 1. Overall, the most common causes of total carcass condemnation were diffuse peritonitis (*n* = 796) and the presence of abscesses (*n* = 527). Notably, a relevant portion of total carcass condemnations was due to the slaughter chain faults (*n* = 145), causing skin burning or over-scalding.

Deaths during transportation and in resting pens at the abattoir were significantly higher in both summer and autumn (*p* < 0.01) (see Appendix A). Carcass condemnation showed a seasonal pattern for all causes except for “other causes”, as recorded at PMI. Peritonitis, abscesses, erysipelas, slaughtering lesions (*p* < 0.01), and generalized jaundice (*p* < 0.05) were significantly more frequent in winter and/or autumn (*p* < 0.01). A seasonal pattern was also demonstrated for skin lesions (in spring, *p* < 0.01) and generalized disorders (in summer, *p* < 0.01) (see Appendix A). Because of the inherent features of the porcine population under study, it was not possible to assess farm or batch effects, if any.

## 4. Discussion

The results of the present large-scale survey, carried out on a huge number of pigs, are broadly in line with the data currently available in the literature [9,10]. Overall, 0.16% of pigs arriving at the abattoir were considered unfit for slaughtering and/or unsuitable for human consumption. Although this percentage seems very low, the resulting economic losses should not be underestimated. From a rough computation, and considering the current price of Italian heavy pigs (about € 2 per kg), the damage is quantifiable as at least in EUR 1,200,000.

Mortality caused by transportation is regarded as a suitable indicator of animal welfare during such a critical process. In this respect, the present survey further highlights that the transportation of pigs to the slaughterhouse is an extremely critical point, which may jeopardize the success of pig farming. As widely expected, and according to the literature available on this topic, losses resulting from the death of pigs during transportation and on arrival at the slaughterhouse were more frequent in the warmer months of the year, when the temperature is often higher than the physiological range for pigs (>25 °C) [9]. Regulation (EC) No. 1/2005 details rules for the transportation of live animals among EU Member States, aiming at safeguarding animal welfare, preventing injury to, or unnecessary suffering for, the animals [11]. However, transportation of animals still poses several threats to animal welfare, during loading and unloading, as well as during transportation per se in relation to grouping, density, handling, and journey duration. The EU Parliament has set up an ad hoc Committee to improve the Regulation (EC) No. 1/2005 [11]. Reasonably, a shift from the export of live animals to the transportation of meat and carcasses could largely solve such an issue, as already suggested in several countries [12].

In the present survey, peritonitis was the most common cause of whole carcass condemnation at PMI. This is rather surprising and deserves further investigation, mainly in order to determine its etiology, which is crucial for understanding the seasonal pattern of such post-mortem findings. Peritonitis is quite common in pigs aged about 8 weeks, is often associated with pleuritis and pericarditis (so called Glasser’s disease), and may be caused by several bacterial pathogens, such as *Glaesserella parasuis*, *Mycoplasma hyorhinis* and *Streptococcus suis* [13]. On the other hand, the occurrence of peritonitis in market weight pigs seems unlikely, unless it represents the chronic outcome of previous disease conditions. Confirming this last assumption, most cases of peritonitis recorded in the present study were chronic (fibroadhesive peritonitis) and so severe and diffuse that carcass trimming appeared economically unviable. Notably, peritoneal lesions were often associated with splenic torsion and could partially result from a vascular defect.

Nowadays, particular attention should be paid to the occurrence of disseminated abscesses scattered throughout the carcass and viscera. This health issue is of growing importance as it seems closely related to tail biting and it might be further exacerbated after the European banning of tail docking [14,15,16]. Accordingly, the higher prevalence of abscesses in winter might result from a seasonal pattern of tail biting. In this respect, data on the effects of temperature on tail biting are contradictory and deserve further investigation [17].

Jaundice represented the third cause of carcass condemnation, its prevalence being similar to that in previous reports in Italy [9]. Jaundice is a rare finding in pigs and can be occasionally caused by viral infections (e.g., porcine circovirus type 2), bacteria (e.g., *Mycoplasma suis*), parasites (e.g., massive infection by *Ascaris suum*), and chemicals (e.g., copper, mycotoxins etc.) [18,19,20,21]. The etiology of jaundice was not investigated herein, although some cases were observed in pigs affected by severe and chronic hepatic lesions (i.e., liver fibrosis and cirrhosis).

*Erysipelothrix rhusiopathiae* is the causative agent of erysipelas and it has long been recognized as zoonotic, even if rarely reported. Notably, most human infections by *E. rhusiopathiae* result from occupational exposure (e.g., in abattoir workers and veterinarians) after skin injuries, thus stimulating the implementation of appropriate preventive measures [22]. *E. rhusiopathiae* can be frequently isolated from the skin and the spleen of pig carcasses, while it is usually absent in the deep muscle. Despite this, the veterinary judgement of erysipelas-affected pigs is very strict and severe due to the zoonotic potential of *E. rhusiopathiae.* According to Regulation (EU) No. 2019/627, porcine carcasses affected by erysipelas must be destroyed [6]. Moreover, the detection of erysipelas at AMI must defer the slaughtering, although the length of this period is not specified [23]. The present survey confirms that erysipelas is still among the most common causes of whole carcass condemnation, with a distinct seasonal pattern; at the same time, it highlights the frequent failure of AMI to detect typical skin lesions in live animals. The reasons for this failure should be further investigated. Reasonably, the huge number of pigs slaughtered per day, as well as the speed of the entire slaughter chain, may reduce the effectiveness of AMI. Moreover, the presence of dirty pigs is commonly experienced at the resting pens [7] and could make the detection of the typical diamond-shaped lesions even more challenging.

Finally, a small portion of carcasses were condemned due to rare diseases (i.e., neoplasms) with unknown etiology (i.e., porcine dermatitis and nephropathy syndrome, ringworm, pityriasis rosea-like lesions) or having been poorly detailed from a diagnostic point of view. As an example, the bloody appearance of the gastroenteric tract has been generically defined as enteritis/enterorrhagia in the present survey, thus including different disease conditions (e.g., severe inflammation of the gut, gastric ulcer, hemorrhagic bowel syndrome). Porcine dermatitis and nephropathy syndrome (PDNS) has been observed worldwide and it is characterized by systemic vasculitis, most likely due to type III hypersensitivity reactions (i.e., an abnormal immune response, mediated by the formation and deposition of immune complexes). The etiology of PDNS is still debated, while its occurrence is sporadic and of poor economic relevance. However, OVs should be aware of PDNS, as it must be considered among the differential diagnoses of African swine fever [24].

## 5. Conclusions

Overall, the present survey provided a broad and topical view of the condemnation of porcine carcasses in Italy. In our opinion, some relevant considerations arise from the collected data. Firstly, mortality during transport and upon arrival at the slaughterhouse stimulates further assessment of animal welfare during such critical stages. The relatively high and likely growing prevalence of abscesses warrants special attention to this health issue, as well as to related disease conditions (namely, tail biting). Finally, the critical analysis of data underlines the need to further *standardize ante-mortem and post-mortem* judgements, as well as to perform suitable diagnostic investigations. This is often challenging, if not impossible, in high-throughput abattoirs. On the other hand, AMI and PMI performed by skilled veterinarians, along with the possibility of correctly identifying the causes of carcass condemnation (even by means of laboratory investigations whenever needed) are crucial for best exploiting the abattoir as a useful source of epidemiological information.

## Figures and Tables

**Figure 1 vetsci-10-00482-f001:**
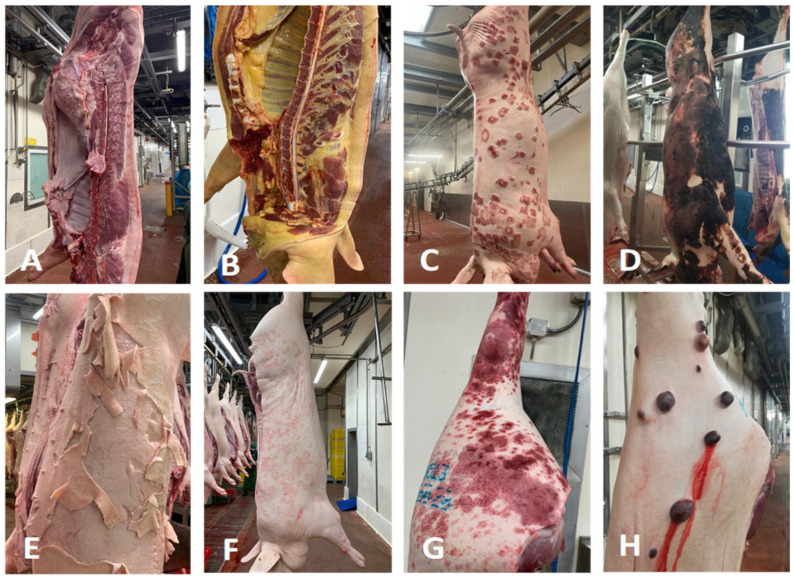
Condemned carcasses. (**A**) A large abscess can be observed in the abdominal cavity, close to the lumbar vertebrae. (**B**) Diffuse and intense yellowish pigmentation of the carcass, resulting from jaundice. (**C**) Erysipelas. Typical raised and diamond-shaped lesions are seen, scattered over the skin. (**D**) Skin burning due to excessive flaming of the carcass. (**E**) External appearance of the carcass due to overscalding, because of too-long duration and/or too hot water in the scalding tank. (**F**) Presence of round-shaped, often coalescing lesions, most likely caused by fungal infection. Pityriasis rosea should be considered as a differential diagnosis. (**G**) Porcine dermatitis and nephropathy syndrome. Multifocal, often coalescing lesions are seen on the caudal surface of the hind leg. Such lesions are slightly raised, well-circumscribed, and dark red in color. (**H**) Skin neoplasms. In this case, cutaneous angiomatosis was diagnosed.

**Table 1 vetsci-10-00482-t001:** Pigs unfit for slaughtering after ante-mortem inspection listed by causes.

Cause	*n*	%
Death during transportation	757	55.58
Death in resting pens at the abattoir	455	33.41
Emergency culling	150	11.01
Total	1362	100.00

**Table 2 vetsci-10-00482-t002:** Carcass condemnation after post-mortem inspection listed by causes.

Cause	*n*	%
Peritonitis	796	39.66
Abscesses	527	26.26
Generalized jaundice	228	11.36
Erysipelas	195	9.72
Slaughtering lesions ^a^	145	7.22
Skin lesions ^b^	56	2.79
Generalized disorders ^c^	28	1.39
Enteritis/enterorrhagia	21	1.05
Other causes ^d^	11	0.55
Total	2007	100.00

^a^ Accidents along the slaughter chain, i.e., overscalding of carcasses while crossing the water tank. ^b^ Ringworm, pityriasis rosea-like lesions, porcine dermatitis and nephropathy syndrome. ^c^ Diffuse and severe pallor (anemia) of skin, skeletal muscle and viscera. ^d^ Neoplastic disorders, sexual odor, acute pneumonia, prominent splenomegaly.

## Data Availability

The dataset used and/or analyzed during the present study is available from the corresponding Author on reasonable request.

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
