# Peer review of "Condemnation of Porcine Carcasses: A Two-Year Long Survey in an Italian High-Throughput Slaughterhouse"

_vetsci, 2023, doi:10.3390/vetsci10070482_

Round 1
Reviewer 1 Report
Major comments
Summary / Abstract
- You should refer clearly to the aim of this study in part ‘’summary’’ as well as in the part ‘’abstract’’.
Introduction
- You should report data about the pig sector in Italy: population, number of slaughtered pigs, mean age and BW of carcasses etc
Materials and Methods
- You should add the approval by an institutional ethical committee.
- Add a section with the records that were investigated and reported to the ‘’results’’ section.
Results
- You should add p-values in Tables 1 and 2.
-
Discussion
- You should underline the importance of results for animal welfare issues at transport and at slaughtering.
Minor comments
- L90: .. slaughtered in Italy). Most of the slaughtered pigs are
- L95: .. information was detailed
- L96: .. and cause of
- L99: .. about the causes of pigs unfit
- L116: .. due to the slaughter
- L117: .. burning or over-scalding
- L164: .. aged about 8 weeks
- L166: .. in market-weight pigs
- L181: by Ascaris suum (italics)
- L185: .. According to Regulation
- L191: .. a small portion of carcasses was condemned
- L198: .. about porcine carcasses of condemnation
Author Response
Response to Reviewer 1 Comments
Major comments
Summary / Abstract
Comment: You should refer clearly to the aim of this study in part ‘’summary’’ as well as in the part ‘’abstract’’.
Response: Summary and abstract have been changed as suggested by the Reviewer.
Introduction
Comment: You should report data about the pig sector in Italy: population, number of slaughtered pigs, mean age and BW of carcasses etc
Response: Actually, such data are reported at the beginning of “materials and methods”. Aiming to avoid repetitions within the manuscript, we would like to maintain the manuscript in the present format. However, if necessary, we are fully available to move such information to the introduction section.
Materials and Methods
Comment: You should add the approval by an institutional ethical committee.
Response: Data has been added at this point.
Comment: Add a section with the records that were investigated and reported to the ‘’results’’ section.
Response: Actually, such data are reported at the beginning of the results section. Please, provide us with more details about still lacking data, to be included within the manuscript.
Results
Comment: You should add p-values in Tables 1 and 2.
Response: p-values cannot be added as no statistical analysis was carried out at this point.
Discussion
Comment: You should underline the importance of results for animal welfare issues at transport and at slaughtering.
Response: Further comments have been added at this point
Minor comments
Response: The text manuscript has been changed/corrected according to the Reviewer’s comments.

Reviewer 2 Report
Very interesting. English editing urgently needed. Usually, in South Africa, all condemned carcasses with lesions likely to be caused by an infectious agent must be sent for laboratory testing. I am sure this happens in Italy as well. Perhaps it is published in another paper. However, it would be good if a table of lab findings could be included for infectious diseases. Erisipelas is a zoonosis and perhaps they should mention safety precautions in place for positive cases.
English language editing required. Perhaps because authors speak Italian; English is second language.
Author Response
Response to Reviewer 2 Comments
Comments: Very interesting. English editing urgently needed. Usually, in South Africa, all condemned carcasses with lesions likely to be caused by an infectious agent must be sent for laboratory testing. I am sure this happens in Italy as well. Perhaps it is published in another paper. However, it would be good if a table of lab findings could be included for infectious diseases. Erisipelas is a zoonosis and perhaps they should mention safety precautions in place for positive cases.
Response: English format has been deeply checked and implemented by Prof. Francesca Rosati, who is now included within the acknowledgments section. A short note has been added about the relevance and significance of erysipelas at slaughter. We appreciate the Reviewer’s comment about the investigative approach to condemned carcasses in South Africa. However, in Italy such microbiological investigations are not mandatory and are not routinely performed, especially when the diagnosis can be effectively made based on gross findings (e.g., PDNS and erysipelas). Therefore, we cannot add further information at this point.
